# How to Test the On-Ice Aerobic Capacity of Speed Skaters? An On-Ice Incremental Skating Test for Young Skaters

**DOI:** 10.3390/ijerph20042995

**Published:** 2023-02-08

**Authors:** Zhenxing Kong, Hanyue Zhang, Mingyue Zhang, Xiao Jia, Jingjing Yu, Junpeng Feng, Shouwei Zhang

**Affiliations:** 1Key Laboratory of Exercise and Physical Fitness, Ministry of Education, Beijing Sport University, Beijing 100084, China; 2School of Physical Education, Northeast Normal University, Changchun 130024, China; 3Department of Kinesiology, College of Human Sciences, Iowa State University, Ames, IA 50011, USA

**Keywords:** speed skating, maximal oxygen uptake, incremental load, test method

## Abstract

Aerobic capacity is important for speed skaters to achieve good results in middle–long distance events. The technical characteristics of speed skating cause intermittent blood flow blockage in the lower limbs. Therefore, an athlete’s aerobic capacity on ice may differ from that measured by cycling or running. Now, the on-ice aerobic capacity lacks methods for conducting aerobic capacity tests on ice. Objective: The objective of this study was to develop a method for measuring on-ice aerobic capacity for young athletes and to compare it with the VO_2max_ test on cycling. Methods: This study established a test method for the on-ice aerobic capacity of young, high-level speed skaters with incremental load (on-ice incremental skating test, OIST) through expert interviews and literature review. In the first part, OIST was used to test the aerobic abilities of 65 youth professional speed skaters (51 males and 14 females) on ice and to explore the correlation with their specific performance. The second part compares the relationship between aerobic capacity on ice and aerobic capacity on bicycle of 18 young high-level male athletes. The third part establishes the regression formula of ice ventilation threshold heart rate. The OIST established in this study can evaluate the on-ice aerobic capacity of athletes from National Level and Level 1&2 in China. The athletes’ on-ice aerobic capacity indicators were significantly lower than those of the cycling test. However, the values of absolute VO_2max_ and absolute ventilatory threshold had a high correlation (R = 0.532, *p* < 0.05; R = 0.584, *p* < 0.05). The regression formula of ventilatory threshold heart rate on ice = 0.921 × HRmax (Cycling test) −9.243. The OIST established in this study meets the characteristics and requirements of the VO_2max_ measurement method. The OIST seems to be able to better evaluate the aerobic capacity of athletes skating on ice. The indicators of maximum oxygen uptake and ventilation threshold in OIST were significantly lower than those in the aerobic cycling test, but there was a good correlation. The aerobic cycling test can be used as an important selection index of the ice aerobic capacity of speed skaters. The regression formula will provide an important basis for coaches to accurately monitor the intensity of ice training.

## 1. Introduction

With the introduction of the Clap Skate in the 1990s, speed skating performance has improved tremendously. For example, the world record for the men’s 10,000 m was 13:30.55 in 1994 before the application of the Clap Skate. However, the current world record is 12:30.74, set at the 2022 Beijing Winter Olympics [1]. There was about a one-minute improvement in the world record over 20 years. De Koning has shown using data modeling that about half of these huge improvements are due to external technological innovations, such as indoor ice rinks and Clap Skate [2]. Furthermore, athletic ability has also improved [3]. Aerobic capacity, as an important ability in speed skating middle-long distance, guarantees excellent competitive performance.

An accurate definition of training intensity is an important criterion for achieving scientific training in training practice and scientific research in speed skating. The joint angles and muscle groups in force generation in cycling are very similar to those in speed skating. Therefore, cycling has become a fundamental part of energy system training for speed skating training [4,5]. The evaluation of aerobic capacity by incremental load experiment and anaerobic capacity by Wingate experiment has become an important means of training intensity development for athletes in speed skating over the years [6,7]. However, the flexion position of skating technical skills can obstruct blood flow to the lower extremities [8,9], affecting the athletes’ aerobic capacity. There may be a significant difference between the athletes’ aerobic capacities on ice and the cycling. Therefore, training intensity monitoring indexes such as target heart rate divided by cycling test results may not be directly used for intensity monitoring of ice training.

It is crucial to evaluate the athletes’ aerobic capacity on ice and its correlation with the aerobic capacity in cycling in the training practice of speed skating. This study developed an on-ice incremental skating test (OIST) for young, high-level speed skaters and compared it with the cycling aerobic capacity test data. The results will help to develop more accurate on-ice training intensity in training practice and contribute to the scientific training of speed skating programs.

## 2. Participants and Methods

### 2.1. Participants

This study recruited 65 youth professional speed skaters from provincial teams in northeastern China for the first part. Among them, 51 were male athletes, and 14 were female athletes. All athletic levels were Level 2 athletes or above. The athletes’ age, heights, and weights are shown in Table 1. Their best results for each distance were retrieved from the website https://www.speedskatingresults.com/ (accessed on 15 November 2022).

This study conducted the second part of the experiment that compared the differences between the OIST and the cycling test. For this part, 18 professional youth speed skating male athletes from northeastern China province teams were recruited, including eight national-level and 10 Level-1 athletes. Four of the 18 athletes completed only the on-ice test, while the remaining 14 completed both the on-ice and cycling test. The basic profiles of the elite male athletes are shown in Table 2. There were no significant differences in age, height, or weight among the different levels of athletes. Athletes who undergo both the cycling test and the on-ice test will undergo the on-ice test first and the cycling test on the third day after the completion of the on-ice test.

All athletes were informed of the study procedures before testing and signed a written informed consent that they confirmed to be free of heavy training, competition, and injury before performing each test.

### 2.2. Design

#### 2.2.1. On-Ice Aerobic Capacity Test

Before the test, we carried out expert interviews and literature combing, and finally decided to take the form of incremental load for the on ice aerobic capacity test. The reference standard for speed was developed based on the latest version of the “Technical Grade Standard for Speed Skaters” of the General Administration of Sports of China. In this study, we established the On-ice incremental skating test (OIST). The details are shown as Table 3.

Test Mode: Continuous incremental loading was used as the loading mode of the protocol for the test. Athletes wore the portable gas metabolic analyzer Cortex Metamax 3B R2 during the whole test and completed the gas collection, and monitored and recorded the changes of heart rate and oxygen uptake values of the subjects throughout the test.

Load intensity: Initial load is 50 s/lap by non-low skating posture to obtain the initial speed; the speed of the first level is divided according to the level of the athlete, different levels of athletes taking a different starting speed; with each level of the single lap time decreasing 1 s/lap, that is, each level of each lap speed compared to the previous level of each lap reducing by the time of 1 s, the movement process allowing athletes the actual speed and rated speed difference within ±1 s. Each level glides for 2 laps. The load intensity settings involved in this study are shown in Table 3.

Test Oval: Testing took place on two regular 400-m indoor lowland long-track ice rinks located in Harbin and Daqing, Heilongjiang, China. The ovals meet the requirements of the International Skating Union. Skaters skated the inner lane of the racecourse (384 m per lap), and the ice temperature and ice surface temperature during the test were in accordance with the requirements of the ISU.

Test Mode: Athletes are required to skate individually using the same skating technique as in the competition. The 1000 m finish line (middle line of the straight) is used as the starting point and the end point of the lap time; the coaches on the sidelines will measure the lap time and ask the athletes to use a good skating posture to complete the test.

Termination Criteria: (1) The Athlete feels exhausted and is unable to continue the exercise. (2) The athlete is unable to reach the rated load at two consecutive levels. (3) The actual speed of a single level at the end of the exercise differs from the rated load by more than 2 s. (4) As the load rises, oxygen uptake plateaus.

#### 2.2.2. Cycling Aerobic Capacity Test

Test Method: Subjects wore a portable gas metabolic analyzer Cortex Metamax 3B R2 for the power bike incremental load experiment, and the changes of heart rate and oxygen uptake values of subjects were monitored and recorded throughout the test. VO_2max_, heart rate, and anaerobic threshold were recorded during the test.

Incremental Load Scheme: The incremental load scheme is shown in Table 4.

Termination Criteria: (1) The athlete feels exhausted and is unable to continue the exercise. (2) The athlete is below 10% of the designated speed for 30 s continuously. (3) A plateau in oxygen uptake as the load rises.

### 2.3. Instruments

MONARK power bicycle (Monark 839E, Monark, Sweden), Portable Gas Metabolic Analyzer (Cortex Metamax 3B R2, Cortex, Leipzig, Germany), Stopwatch (Casio HS-70, Casio, Tokyo, Japan), Polar Heart Rate Sensor (Polar H10, Polar, Kempele, Finland).

### 2.4. Statistical Analyses

IBM SPSS Statistics (version 22.0; IBM Corp, Armonk, NY, USA) performed the statistical analyses. All data obtained from the experiment were presented as mean ± SD. The independent samples *t*-test compared the results of the OIST for male and female athletes. Furthermore, the paired samples *t*-test compared the differences between the on-ice and the cycling test results for the same subjects. When *p* < 0.05, the statistical differences were significant, and when *p* < 0.01, the statistical differences were highly significant. Pearson correlation analyzed the correlation between OIST result vs. Personal Best in different distances and the results of the two testing methods. The correlation coefficients were classified as low correlation (0.1–0.3), moderate correlation (0.3–0.5), high correlation (0.5–0.7), and very high correlation (0.7–0.9). When *p* < 0.05, the statistical differences were significant. Simple linear regression was used to establish the regression equation for the ventilatory threshold heart rate on ice. Durbin–Watson was used to test the independence between variables.

## 3. Results

### 3.1. On-Ice Aerobic Testing for Different Sexes Athletes

In this study, there were significant differences in oxygen uptake on maximum value and ventilatory threshold (*p* < 0.01) between male and female athletes. There were no significant differences in the speed on maximal oxygen uptake, speed on ventilatory threshold, maximal heart rate, and ventilatory threshold heart rate (*p* > 0.05) among the athletes of different genders in the OIST (Table 5).

This study explored the correlation between the relevant indicators measured by OIST and the athletes’ personal best performance in the 1500 m, 3000 m, and 5000 m (Table 6). The maximum oxygen uptake speed measured by OIST, speed on ventilatory threshold, and relative value of VO_2max_ showed a very high and significant correlation (R = 0.805, *p* < 0.01), high correlation (R = 0.672, *p* < 0.01), and moderate correlation (R = −0.354, *p* < 0.05), respectively, with the athletes’ personal best in 5000 m. The absolute value of VO_2max_ and the absolute and relative values of the anaerobic threshold were insignificantly correlated with the 5000 m personal best.

Relative VO_2max_, speed on maximal oxygen uptake, relative ventilatory threshold, and speed on ventilatory threshold showed a very high correlation with the athletes’ personal best in the 3000 m. The absolute value of VO_2max_ and the absolute value of the ventilatory threshold highly correlated with the personal best in the 3000 m. Furthermore, the absolute and relative VO_2max_, speed on VO_2max_, absolute, relative ventilation threshold, and speed on ventilation were very significantly and highly correlated with performance in the 1500 m.

### 3.2. On-Ice Aerobic Testing for Elite Male Athletes of Different Levels

This study compared the data of elite male athletes of different levels performing the OIST established in this study (Table 7). The comparison showed no significant difference between the national-level and the Level-1 athletes in the values of absolute maximal oxygen uptake (VO_2max_), relative maximal oxygen uptake (VO_2max_), maximum heart rate (HR_max_), total test duration, respiratory quotient (RER), absolute ventilatory threshold (VT), and relative ventilatory threshold (VT) (*p* > 0.05). There were significant differences in the speed on maximum oxygen uptake rate (V VO_2max_) (*p* = 0.028 < 0.05), HR on ventilatory threshold (HR_VT_) (*p* = 0.000 < 0.01), and speed on ventilatory threshold (V VT) (*p* = 0.004 < 0.01) between the different levels of athletes.

### 3.3. Comparison of Cycling Aerobic Test and On-Ice Aerobic Tests

This study conducted paired *t*-tests on two test indicators for the 14 athletes who completed both cycling and OIST tests (Table 8). The results showed that there were highly significant differences in absolute VO_2max_, relative VO_2max_, maximum heart rate (HR_max_), absolute ventilation threshold (VT), relative ventilation threshold (VT), ventilation threshold as a percentage of maximum oxygen uptake (VT/VO_2max_), and ventilation threshold heart rate (HR_VT_) between the two tests (*p* < 0.01). There were no significant differences in respiratory quotient (RER) and maximum heart rate (HR_max_) (*p* > 0.05).

This study conducted Pearson correlation analysis on the results of the same indexes tested by cycling and ice aerobic tests (Table 9). There was a moderate correlation between absolute maximal oxygen uptake and absolute ventilatory threshold (R = 0.532, *p* < 0.05; R = 0.584, *p* < 0.05) and a strong correlation with maximum heart rate (R = 0.754, *p* < 0.01). Except for the percentage of maximal oxygen uptake at the ventilatory threshold, there was a weak and insignificant correlation between the other four indices (*p* > 0.05).

### 3.4. Establishment of On-Ice Ventilation Threshold Heart Rate Speculation Formula

This study combined the results of Section 3.1 and Section 3.2 to establish the speculation of on-ice ventilation threshold heart rate. It was based on the maximum heart rate of the cycling due to the complexity of the on-ice testing process.

The Durbin–Watson test result was close to 2, indicating that the observations in this regression were independent (Table 10).

Regression model: HR for On-Ice Ventilation Threshold = $0.921\times HR_{max}\left(Cycling\;test\right)-9.243$.

## 4. Discussion

The cycling incremental load test has been considered an effective means of testing and evaluating the aerobic capacity of speed skaters since the 1980s. Various condition factors influence ice testing. Therefore, testing and evaluating aerobic capacity on ice has been a research area for many years. This study established a method for independently testing aerobic capacity on ice with incremental load by combining the results of preliminary expert interviews and literature combing. After testing and comparison, the method showed good validity in the aerobic capacity on-ice evaluation. Furthermore, this study investigated the differences and correlations between the cycling and ice aerobic capacity tests and discovered that both tests had good correlations in three indexes: absolute maximum oxygen uptake, absolute ventilation threshold, and maximum heart rate. This study also established a method to infer the maximum heart rate on ice from the cycling test and a regression formula to infer the maximal heart rate of the cycling test from the ventilation threshold heart rate on ice.

Hill et al., introduced the maximum oxygen uptake (VO_2max_) in 1923. Related research has quickly become a hot topic of research in the field of exercise for training and health. The selection of test methods, determination indexes, and criteria for VO_2max_ has been an important part of the research in this field. The direct test method of maximal oxygen uptake includes two main forms: (i) constant power and (ii) incremental load, with the incremental load being the common form [10]. The study by M. Leone and John J. Durocher [11] showed significant variability in the aerobic capacity indicators tested on the ice and in the laboratory for ice hockey players. It suggested the importance of specific aerobic capacity testing. Based on preliminary expert interviews and literature analysis, this study chose an incremental load protocol similar to Petrella’s [12] on-ice test for hockey players. It involved a movement pattern that required athletes to follow the same technical skills as in competition and training, with a high consistency degree with training conditions.

From the test protocol design perspective, the OIST test protocol met the basic requirements of the VO_2max_ test protocol. It required the participation of large muscle groups, a test time of six minutes or more, and a maximum oxygen uptake of around 12 to 20 min. The oxygen uptake plateau is the gold standard for determining maximum oxygen uptake [13], but related studies have shown that the oxygen uptake plateau percentage was low in the actual test. Therefore, respiratory exchange ratio (RER), maximum heart rate (HR_max_), and blood lactate concentration (BLA) became the secondary indexes for better determination of the maximum oxygen uptake.

The criteria for determining the maximum heart rate were based on the predicted maximum heart rate (220–age), the maximum heart rate during exercise ≤ (predicted maximum heart rate—10), or the maximum heart rate during exercise ≥ 95% of the predicted maximum heart rate. The blood lactate concentration criteria were BLA ≥ 8 mmol/L. The criteria for determining the respiratory quotient were RER > 1.10 or 1.15 [14,15]. This study’s OIST was based on the maximum heart rate and respiratory quotient. Furthermore, it also met the maximum oxygen uptake test criteria in terms of maximum heart rate, respiratory quotient, and other metabolism-related indicators.

This study used OIST to test high-level young male speed skaters at two levels: a national athlete level and Level 1 athlete level. Despite the different levels, no significant differences were found in the maximal oxygen uptake and ventilation threshold indexes obtained by the aerobic capacity test on ice. It indicated that the method established in this study had good generalizability. Testing athletes of different genders and levels showed that the indicators measured by OIST correlated well with the athletes’ best performance in middle-long distances, indicating the advantages of OIST in evaluating specific abilities.

OIST took speed skating special techniques into consideration, unlike the cycling measurement method, which was better for evaluating special aerobic capacity. This study also found a significant difference in the maximal oxygen uptake rate of athletes of different levels (*p* = 0.028 < 0.05), indicating that OIST had some advantages in differentiating the special competitive level.

Cycling training is an important part of the speed skating program training. The cycling test for aerobic capacity measurement had good significance in evaluating and developing intensity for cycling in speed skating training. This study found that OIST and cycling tests could make athletes reach force exhaustion, develop maximal heart rate, and achieve a high respiratory quotient. However, the absolute VO_2max_, relative VO_2max_, VO_2max_ velocity, absolute anaerobic threshold oxygen uptake, relative anaerobic threshold oxygen uptake, and anaerobic threshold heart rate measured by OIST were significantly lower than those measured by the cycling test.

The movement patterns of speed skating events differ vastly from those of other events. For example, during the running process, the pedaling action generates a force opposite to the direction of movement, pushing the body in the direction of motion [16,17,18]. However, in speed skating, the skater’s blade is at a right angle to the skating direction due to the special characteristics of the equipment and ice surface. The skater pushes the blade sideways to obtain forward speed by force. With the development of modern speed skating techniques, the athlete must keep the torso almost horizontal during the skating process. It minimizes the frontal area and reduces air friction losses [19]. The knee angle significantly affects skating efficiency. Research shows that peak skating power can be reached at a knee angle of 130° [20,21]. The flexion position of the hip and knee joints causes the lower limbs to be in an intermittent ischemic process during skating [22]. Therefore, the oxygen supply to the lower limbs is greatly restricted during skating.

Foster et al. [8] and Koepp et al. [23] showed that the absolute values of maximal oxygen uptake were in the following order for different testing methods: cycling > treadmill low walking > roller skating > ice skating. This study’s testing results showed the same trend, but OIST caused a higher decline. Van Ingen Schenau’s [18] study on adult elite-level athletes showed that VO_2max_ for cycling and 3000 m on-ice full power skating were 64.6 ± 3.5 mL/kg/min and 59.4 ± 3.7 mL/kg/min, respectively, with a decrease of about 8%. This study showed a decrease of about 16%, indicating that the skating technique of youth athletes and the skating economic efficiency need further improvement.

The cycling test was simpler to perform than the OIST test. Therefore, this study investigated the correlation between OIST and cycling aerobic capacity test and found a highly significant correlation between OIST and cycling test in the absolute and relative values of VO_2max_ and anaerobic threshold. In addition, OIST showed a better evaluation of the aerobic capacity of athletes. The cycling test using a cycle ergometer could select athletes with good aerobic capacity on the ice during the young athletes’ selection process. Therefore, the cycling test can be used to evaluate aerobic capacity and aerobic capacity on ice.

Finding the optimal training intensity distribution is key to enhancing an athlete’s performance. Two main patterns of training intensity distribution may occur in endurance athletes: the polarization pattern and the threshold pattern [9,24,25]. The lactate threshold/ventilation threshold is an important index to classify exercise intensity for the threshold model. Combining the findings from the first and second parts of this study found no significant difference in maximum heart rate between athletes performing the cycling and ice tests. However, there was a highly significant difference in ventilation threshold heart rate (*p* < 0.01), with a difference of approximately 15 b/min. Lower ventilation threshold on ice involved factors such as skating posture, temperature, and reduced blood oxygen levels in the exercising muscle groups. This finding will guide the ventilation threshold intensity for training on ice. Furthermore, the ventilation threshold heart rate obtained from cycling tests cannot be used directly to guide threshold training on ice in training practice.

This study established a speculative formula for the ventilation threshold heart rate on ice with the maximum heart rate of the cycling test as the independent variable. It improved the accuracy of monitoring the intensity of on-ice aerobic training in training practice and will guide coaches to monitor exercise intensity through heart rate in training practice.

Limitation: Since the athletes recruited in this study are all young athletes whose technical and aerobic levels are still in the development stage, the relevant conclusions of this study may not apply to elite athletes. We will also continue to conduct research to explore the characteristics of aerobic capacity of elite athletes on ice.

## 5. Conclusions

The OIST established in this study meets the characteristics and requirements of the maximal oxygen uptake test method. Evaluation of different levels of athletes and different testing methods found that OIST can evaluate the aerobic capacity of athletes skating on ice.The peak maximal oxygen uptake and ventilation threshold-related indexes of the athlete on-ice aerobic test were significantly lower than those of the cycling aerobic test but had a good correlation. The cycling test can be an important selection index for speed skaters’ on-ice aerobic capacity.There are significant differences between on-ice and cycling ventilation threshold heart rates. This study established a regression formula for on-ice ventilation threshold heart rate with cycling test maximum heart rate as the independent variable. The findings will provide a basis for coaches to accurately monitor on-ice training intensity.

## Figures and Tables

**Table 1 ijerph-20-02995-t001:** Basic Profiles of All Athletes.

Gender	*n*	Age (y)	Height (cm)	Body Weight (kg)
Male	51	15.97 ± 1.33	171.38 ± 24.08	61.90 ± 8.01
Female	14	15.14 ± 1.26	165.49 ± 5.66	54.43 ± 5.05
Total	65	15.80 ± 1.35	170.17 ± 21.69	60.36 ± 8.06

**Table 2 ijerph-20-02995-t002:** Basic Profiles for Elite Male Athletes.

Category	*n*	Age (y)	Height (cm)	Body Weight (kg)
National level	8	16.54 ± 1.21	173.63 ± 1.85	62.49 ± 3.11
Level 1	10	15.90 ± 1.24	175.05 ± 4.44	63.08 ± 6.83

**Table 3 ijerph-20-02995-t003:** OIST Load Intensity.

	Male, National Level	Male, Level 1	Male, Level 2	Female, National Level	Female, Level 1	Female, Level 2
Preparation	50 s/lap	50 s/lap	50 s/lap	50 s/lap	50 s/lap	50 s/lap
Level 1	41 s/lap	42 s/lap	43 s/lap	42 s/lap	43 s/lap	44 s/lap
Level 2	40 s/lap	41 s/lap	42 s/lap	41 s/lap	42 s/lap	43 s/lap
Level 3	39 s/lap	40 s/lap	41 s/lap	40 s/lap	41 s/lap	42 s/lap
Level 4	38 s/lap	39 s/lap	40 s/lap	39 s/lap	40 s/lap	41 s/lap
Level 5	37 s/lap	38 s/lap	39 s/lap	38 s/lap	39 s/lap	40 s/lap
…	…	…	…	…	…	…

**Table 4 ijerph-20-02995-t004:** The Cycling Incremental Load Scheme.

Level	Power (W)	Duration (min)	Rpm (r/min)
Rest	0	1	0
Level 1	0	1	70
Level 2	150	3	70
Level 3	200	3	70
Level 4	250	3	70
Level 5	300	3	70
…	…	3	70
Recovery	50	3	/

**Table 5 ijerph-20-02995-t005:** On-Ice Aerobic Testing Result for All Participating Athletes (*n* = 65).

Indicator	Male (*n* = 51)	Female (*n* = 14)
Absolute VO_2max_ (L/min)	3.16 ± 0.37	2.37 ± 0.18 **
RelativeVO_2max_ (mL/kg/min)	51.50 ± 5.19	44.10 ± 4.51 **
V VO_2max_ (s/lap)	37.44 ± 2.12	38.41 ± 2.57
HRmax	192.86 ± 7.89	189.50 ± 11.75
Absolute VT (L/min)	2.57 ± 0.32	2.06 ± 0.23 **
Relative VT (mL/kg/min)	41.85 ± 4.03	37.93 ± 4.57 **
VT/VO_2max_ (%)	89.83 ± 4.81	84.02 ± 6.21
V VT(s/lap)	41.33 ± 1.78	42.29 ± 2.64
HR_VT_	177.30 ± 12.75	168.80 ± 15.18

Note: ** *p* < 0.01.

**Table 6 ijerph-20-02995-t006:** Correlation between OIST and Personal Best in 1500 m, 3000 m, and 5000 m.

Indicator	1500 m (*n* = 54)	3000 m (*n* = 27)	5000 m (*n* = 33)
Absolute VO_2max_ (L/min)	−0.670 **	−0.638 **	−0.261
RelativeVO_2max_ (mL/kg/min)	−0.583 **	−0.715 **	−0.354 *
V VO_2max_ (s/lap)	0.660 **	0.719 **	0.805 **
Absolute VT (L/min)	−0.647 **	−0.639 **	−0.190
Relative VT (mL/kg/min)	−0.592 **	−0.763 **	−0.291
V VT(s/lap)	0.683 **	0.737 **	0.672 **

Note: * *p* < 0.05, ** *p* < 0.01.

**Table 7 ijerph-20-02995-t007:** On-Ice Aerobic Test Results for Different Levels of Elite Male Athletes.

Indicator	National-Level (*n* = 8)	Level-1 (*n* = 10)	*p*-Value
Absolute VO_2max_ (L/min)	3.40 ± 0.22	3.39 ± 0.34	0.961
Relative VO_2max_ (mL/kg/min)	54.50 ± 3.16	54.00 ± 4.14	0.782
HR_max_ (b/min)	198.36 ± 6.48	194.60 ± 6.80	0.250
V VO_2max_ (s/lap)	34.88 ± 1.17	36.33 ± 1.34 *	0.028
Total Test Duration (s)	848.38 ± 87.02	821.50 ± 87.80	0.526
RER	1.16 ± 0.07	1.11 ± 0.05	0.081
Absolute VT (L/min)	2.74 ± 0.32	2.72 ± 0.28	0.925
Relative VT (mL/kg/min)	43.88 ± 3.14	43.500 ± 2.59	0.785
VT/VO_2max_ (%)	80.88 ± 8.53	80.58 ± 2.96	0.918
HR_VT_ (b/min)	188.14 ± 8.08	172.71 ± 9.84 **	0.000
V VT (s/lap)	38.88 ± 1.46	41.10 ± 1.37 **	0.004

Note: * *p* < 0.05, ** *p* < 0.01.

**Table 8 ijerph-20-02995-t008:** Results from Cycling Aerobic Test and OIST (*n* = 14).

Indicator	Cycling Aerobic Test	OIST	*p*-Value
Absolute VO_2max_ (L/min)	4.06 ± 0.59	3.41 ± 0.32 **	0.000
Relative VO_2max_ (mL/kg/min).	65.07 ± 8.65	54.64 ± 3.93 **	0.000
HR_max_ (b/min)	197.5 ± 7.75	195.64 ± 6.48	0.198
RER	1.06 ± 0.06	1.12 ± 0.06	0.943
Absolute VT (L/min)	3.62 ± 0.55	2.75 ± 0.30 **	0.000
Relative VT (mL/kg/min)	57.93 ± 7.69	44.00 ± 2.83 **	0.000
VT/VO_2max_ (%)	89.48 ± 6.57	80.69 ± 6.35 **	0.004
HR_VT_	188.14 ± 8.08	172.71 ± 9.84 **	0.000

Note: ** *p* < 0.01.

**Table 9 ijerph-20-02995-t009:** Correlation analysis of the Results from the Cycling Aerobic Test and OIST (*n* = 14).

Indicator	Correlation Coefficient	*p*-Value
Absolute VO_2max_ (L/min)	0.532 *	0.050
Relative VO_2max_ (mL/kg/min)	0.423	0.131
HR_max_ (b/min)	0.754 **	0.002
RER	0.584 *	0.028
Absolute VT (L/min)	0.357	0.210
Relative VT (mL/kg/min)	−0.029	0.922
VT/VO_2max_ (%)	0.388	0.171

Note: * *p* < 0.05, ** *p* < 0.01.

**Table 10 ijerph-20-02995-t010:** Model Fundamentals.

R	R^2^	Adjusted R^2^	Durbin-Watson
0.726	0.527	0.488	1.907

## Data Availability

The datasets used and/or analyzed during the current study are available from the corresponding author on reasonable request.

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
