# Peer review of "How to Test the On-Ice Aerobic Capacity of Speed Skaters? An On-Ice Incremental Skating Test for Young Skaters"

_ijerph, 2023, doi:10.3390/ijerph20042995_

Round 1

Reviewer 1 Report

Introduction

No comments

Methods

lines 68-69: please provide citation for the first part of the experiment

lines 68-74: Why did so few of the initial 65 athletes complete these tests? 

lines 71-73: Why did only 14 out of 18 complete both tests? 

lines 97-98: Please clarify this: "the movement process allows athletes the actual speed and rated speed difference within ± 1s"

lines 106-109: Was each athlete tested individually on the ice-track, or did they skate in pairs or groups?

line 113: Were the researchers able to monitor the oxygen uptake in real time?

lines 131: When you write the t-test observed, do you mean a t-test was used to compare results between two groups? What two groups? It says later in that line, "all athletes and elite male athletes," but earlier in the methods it indicates that the sample was from only elite male athletes, and divided into national level and level 1 rated skaters.

lines 133-134 and lines 138-139: It is wrong to call a p value < 0.01 "highly significant;" if you want to discuss the magnitude of difference, please use an appropriate effect size analysis

Results

line 142: What groups are being compared in this sentence?

lines 143-145: These do not exactly match the results presented in table 5

lines 143-144: When you write "speed of maximal oxygen uptake," and "speed of ventilatory threshold," do you mean skating speed when those physiological points were reached? Applies to lines 150 and throughout the manuscript as well

Table 5: What does the V in V Vo2max and V VT stand for? 

table 5: related to comment about line 131: it is unclear in the methods who actually performed what tests

lines 150-151: Here you wrote "maximal oxygen uptake speed," instead of "speed of maximal oxygen uptake," which I assume is meant to denote the same variable, but especially with the phrasing of "maximal oxygen uptake speed," it makes it seem like what you were measuring was how fast they could achieve their VO2max, when as I mentioned in a previous comment, I believe you mean the skating speed achieved when the athlete reached their VO2max

Table 6: Please continue to put the units of measure next to each variable

line 166: What does the word established refer to here? Was this sentence not completed? 

lines 171-171: It is unnecessary to write <0.05 and <0.01 since you reported the exact p value

lines 188-190: Based on table 9, percentage of maximal oxygen uptake at the ventilatory threshold also seems to not be significantly correlated between the on ice and cycling tests

section 3.4, lines 193-201: First, this was not indicated in the methods. Second, based on the results, which indicated significant differences between the cycling and on-ice tests for all but two of the cardiovascular response variables, why base it on the cycling instead of the on-ice test? Especially when HR at VT was specifically one of the variables that was significantly different between the two tests, this doesn't seem like a logical decision to treat the test results interchangeably. 

lines 195-196: It seems like maximum HR on cycling test was chosen a priori; why use that variable instead of any other variable? Or why not use a regression analysis procedure to indicate which variable was most predictive of HR at VT during the on-ice test?

Discussion

line 209: This research did not establish reliability (as  far as described, only a single trial was performed)

line 240: Why are you bringing up blood lactate? That was not measured in this study (or at least not reported).

line 245: "used OIST to test high-level young male speed skaters"--again, there are discrepancies throughout this manuscript about who actually is included in the sample

line 254: What do you mean by "special aerobic capacity?"

line 267: did you mean to write "running"? Or is "pedaling" in the next sentence incorrect? 

lines 266-277: This seems tangential to the reported research

line 281: A higher decline than what? And, when was a decline in oxygen uptake measured?

lines 289-290: How are you making the judgment that the OIST showed a better evaluation of the aerobic capacity of athletes?

line 290: What do you mean "using a power cycling?" 

General: This discussion section does not provide a discussion of limitations

Conclusion

line 315: "Can better evaluate" compared to what? How are you making this judgement? 

Author Response

Dear reviewer

Thank you for your comprehensive, detailed and professional advice. We benefit a lot from your valuable advice. Thanks for your hard work, we have also made the following modifications according to your comments:

Reviewer 2 Report

Comments to authors

Manuscript ID: ijerph-2122045

Title: How to Test the On-Ice Aerobic Capacity of Speed Skaters? - A On-Ice Incremental Skating Test for Young Skaters

 ___________________________________________________________________

General

Thank you for inviting me to review this paper. When it comes to the topic, this study provides some new insights and the content is within the scope of the journal and the special issue “Winter Sports Implications for Training, Environmental and Health”. However, there are parts of the Methods that are unclear and no specific aim is provided in the Abstract nor the Introduction and no hypothesis or research questions. In addition, I think that the rationale for this study is not well founded in the Introduction section. In addition, the Discussion section lacks argument considering method limitations. Lastly, an expert in the English language should review this paper. It has some English language mistakes and suffers from lack of clarity. Past and present tense are mixed in some parts and “subjects” and “participants” and “skaters” are used interchangeably. There are also many dogmatic statements and more humble expressions and discussions are anticipated. Needless to say, this manuscript needs a major re-work and I suggests that it will not be accepted for publication in its current condition.

 Abstract

Needs to be re-written to clarify rationale, aim, methods and results. Please re-write in correct tense and order throughout the Abstract.

 Specific comments

Please provide the aim of the study.

Add information about the number subjects in the Methods.

Provide correct Methods used (design and procedure).

Page 1, line 15: “Therefore, the on-ice aerobic capacity lacks methods for conducting aerobic 15 capacity tests on ice.”. What do the authors mean by this? There is no rationale given for the word “therefore”.

Page 1, line 20: “from different levels”. Please state this levels.

Page 1, line 22: “had a high correlation”. Please provide the r-value and p-value.

Page 1, line 25: “The OIST can better evaluate the aerobic capacity of athletes skating on ice.” Compare to what? Better than what? Please, also use a more humble statement such as “seems to”.

Page 1, line 28: “The regression formula will provide…”. Dogmatic statement. Also, in the main text, there is a lack of information regarding the regression formula. It is not described in the Methods.

 Introduction

 General comments

This section needs re-work to better clarify and emphasize the aim and the rationale of this study and an aim is not provided, nor hypothesis or research question. In addition, several statements lacks references. In addition, there is no clarification considering the 2 different samples in the study? Why did the author use two groups of youth professional skaters? Please develop and clarify.

 Specific comments

Page 1, Line 38: ”De Koning has shown using data modeling that about half of these huge improvements are due to external technological innovations, such as indoor ice rinks and Clap Skate.” Please provide reference.

Page 1, line 41-42: “Aerobic capacity, as an essential ability in speed skating middle-long distance, guarantees excellent competitive performance”. Provide references.

Page 2, Line 53-54: ”It is crucial to evaluate the athletes’ aerobic capacity on ice and its correlation with the aerobic capacity in cycling in the training practice of speed skating.” Crucial is a very strong word. The authors need to better clarify why it is crucial to evaluate the correlation to cycling. The previous argument in the Indtroduction do not support this statement.

Materials and Methods

 General comments

The Method section lacks information regarding study design, recruitment process, inclusion and exclusion criteria, procedure and statistical power (sample size calculation). Also, there are some repeated, duplicated information.

Procedure is lacking for the test session. Where the tests performed on the same day? If not, how many days apart? If tested on the same day, what was the amount of rest between the tests? In what order were the tests performed? In addition, how was the skaters recruited? How many were invited and how many declined participation? What were the inclusion criteria and what were the exclusion criteria? How many were excluded and due to what exclusion criterion? On what basis did the authors recruit 65 skaters? Sample size calculation and based on what variable?

Also, the group affiliation name “All athletes” are somewhat confusing and aiming to all athletes, including the second part subjects as well. In addition, the author state that both these groups consists of professional youth athletes, but only refers to the second group to elite athletes. However, if you are a professional athlete you must be consider as an elite athlete. Also, the Elite Male Athlete group is in turn divided in 2 groups with n=8 and n=10. This small sample must be seen as a limitation. Lastly, there is an additional group consisting of 46 athletes from the group of the 18 elite athletes. This does not add up and needs to be better explained.

Specific comments

Page 2, line 71-72: Then, 14 athletes completed the cycling test; 18 completed the OIST test, and 14 completed both the cycling and OIST tests. This does not add up. 14+18+14 is 46 which is far more than the recruited sample of 18.

Page 2, line 77-79: “All athletes were informed of the study procedures before testing and signed a written informed consent that they confirmed to be free of heavy training, competition, and injury before performing each test.” What type of injury? All injuries? What are the definition of “heavy training”? In addition, did the subjects not provide an informed consent to participate?

Page 4, line 131: ”The independent samples t-test observed the results...”. The independent samples t-test is a test that analyze differences, thus the test is used to compare not to observe.

Page 4, line 134-136: “Pearson correlation analyzed the correlation between OIST result vs. Personal Best in different distances and the results of the two testing methods.” I do not fully understand this sentence, please clarify.

Page 4, line 138-139: ”When p < 0.05, the statistical differences were significant, and when p < 0.01, the statistical differences were highly significant.” The authors have already provide this information above in the Statistical method section. Also, the previous sentence refers to correlation, and thus, this sentence is incorrect at this point (since it refers to differences).

Results

 General comments

Because of the errors and confusing information in the Method section, I have trouble fully understand this section. In addition, please provide specific p-value for each test or use the ≥ or ≤ when aiming at several p-values, stating the lowest respectively the highest p-value. The section about the correlations need to be re-written. The authors provide the R-value both in the text and in the Table 6. However, the tables should add more information, complete the text, and not duplicate. I suggest the author state the specific p-value in the text and the degree of correlation according to the classification, but not the actual R-value (since this is already presented in the Table). In addition, the authors provide many results using abbreviations. This abbreviations; Absolute VO2max, Relative VO2max, HRmax, V VO2max, RER, Absolute VT, Relative VT, VT/VO2max, HRVT and V VT, should all be described in and spelled out in the Method section. It must be clear in the Methods all the variables that the study aims to investigate.

 Specific comments

Page 4, line 142: – “There were significant differences in maximal oxygen uptake and ventilatory threshold (p < 0.01)”. Compared to what? Or do the authors mean between maximal oxygen uptake and ventilatory threshold? Please clarify and also see comments above considering the p-value.

Page 4, line 143-145: ”No significant differences in the speed of maximal oxygen uptake, speed of ventilatory threshold, maximal heart rate, and ventilatory threshold heart rate (p > 0.05) among the athletes of different genders in the OIST”. Differences between what or compared to what? Also see comments above considering the p-value.

Table 5: Please add total N in the heading.

Page 5, line 155 – “...personal best”. I still do not fully understand how the personal best value was achieved.

Page 5, line 156-161. Please add specific p-value for your result statements.

Table 6: What is the n=54 coming from? How can there be 54 skaters? Also, it should state correlations and not correlation since it several correlations provided. Add, heading stating r-value so the author can see that this is Pearson’s you are referring to (or correlation coefficient as in Table 9). Also add what the * and ** represents below the Table (as in Table 5). Add this information to every table that provide * and **. In addition, please see comment in the general comment section above.

Page 5, section 3.2: The author do not need to describe that p=0.028 is less than 0.05. Please remove these parts from this section.

Table 7: The HRVT is not described in the main text. Also, what is Total Test Duration and why has this been analyzed?

Page 6, section 3.3: As mentioned in the comments to the Method section, I do not understand what this group of 14 athletes are coming from.

Page 6, line 184: “This study conducted Pearson correlation analysis on the results of the same indexes tested by the two methods of cycling and ice aerobic tests.” What? I do not understand this sentence. This sentence also consist of Method description, which need not be repeated.

Page 6, section 3.4. Establishment of On-Ice Ventilation Threshold Heart Rate Speculation Formula: This needs to be described in the statistical method section.

 Discussion

 General comments

This section also need a major rework. The Discussion section are too weak and it lacks of interpretation of data. All results must be discussed. I also suggest that the Discussion starts with the major result, main finding, of the present study. In addition, I suggest that the authors discuss more thorough the significance of their results and provide a broader discussion in relation to other studies result together with limitations and generalization. The Discussion section lacks arguments regarding the methodological limitations of the study.

 Conclusions

 General comments

The conclusions drawn need to be revised and should directly respond to the purpose of the study and the research questions. If the aim was to establish a regression formula, this need to be emphasize in the Introduction, stated as an aim, and described in the Methods.  

 Specific comments

Page 9, line 323-325: “The findings will provide an important basis for coaches to accurately monitor on-ice training intensity.” Very strong claims. Please be more humble in your statements.

 References

The references seem to be up to date, but as stated above, the authors must provide more references to their arguments. The manuscript are lacking references to support some arguments.

 Tables

Please find comments above.

Author Response

Dear reviewer

Thank you for your comprehensive, detailed and professional advice. We benefit a lot from your valuable advice. We still need to make progress in English literature writing, and we are very sorry that we did not express clearly. Thanks for your hard work, we have also made the following modifications according to your comments:

Reviewer 3 Report

A very accurate and valuable article with great potential for practical applications, but it requires a few corrections and additional explanations.

1. Putting a question in the title and double-wording is a bad manner inconsistent with the methodology of scientific work. The maturity of the researchers is also expressed in the ability to construct the title as a one-sentence dilemma.

2. The division of the study groups is unclear. It is not entirely clear whether women or only men were tested on the bicycle ergometer. This is essential when interpreting the results.

3. The study protocol and procedure should be discussed in more detail. During the OIST study, were ventilation analyzers placed on the players' bodies and what was their weight?

4. Avoid repeating the text from the introduction and the results in the discussion.

5. The request for a better assessment of the aerobic capacity of skaters by OIST (it is not known in relation to what other measurement) is unjustified.

Author Response

(The authors gave the same response as above.)

Round 2

Reviewer 2 Report

Comments to authors -revised version

Manuscript ID: ijerph-2122045

Title: How to Test the On-Ice Aerobic Capacity of Speed Skaters? - A On-Ice Incremental Skating Test for Young Skaters

_____________________________________________________________________________

General

I find the paper improved and that some of my comments have been addressed appropriate. However, there are still some parts that needs to be revised. In addition, I strongly suggests an expert in the English language should review this paper.

Introduction

Please address comments below.

Specific comments

Page 1, line 49: “Aerobic capacity, as an essential ability in speed skating middle-long distance, guarantees excellent competitive performance”. Provide references.

Page 61: ”It is crucial to evaluate the athletes’ aerobic capacity on ice and its correlation with the aerobic capacity in cycling in the training practice of speed skating.” Crucial is a very strong word. The authors need to better clarify why it is crucial to evaluate the correlation to cycling. The previous argument in the Indtroduction do not support this statement.

Methods

General comments

The Method section still lacks information regarding study design and recruitment process, inclusion and exclusion criteria, procedure and statistical power (sample size calculation).

Please state the study design. I believe that it should be cross-sectional?

Please also address the following: How many were invited and how many declined participation? What were the inclusion criteria and what were the exclusion criteria? How many were excluded and due to what exclusion criterion? On what basis did the authors recruit 65 skaters? Sample size calculation and based on what variable?

Specific comments

Line  72: ”...were from the website...? Do the authors mean ”..were retrieved from the website...?

89-91: “All athletes were informed of the study procedures before testing and signed a written informed consent that they confirmed to be free of heavy training, competition, and injury before performing each test.” What type of injury? All injuries? What are the definition of “heavy training”? In addition, did the subjects not provide an informed consent to participate?

Page 4, line 131: ”The independent samples t-test observed the results...”. The independent samples t-test is a test that analyze differences, thus the test is used to compare not to observe.

Results

Please address comments below.

Specific comments

Line 157-158: – “For the young male and female athletes in this study, there were significant  differences in oxygen uptake on maximum valuemaximal oxygen uptake and ventilatory threshold (p < 0.01).”. Compared to what? Or do the authors mean between gender? Please clarify and also see comments above considering the p-value. In addition, I think sexes is the correct term instead of gender (Sex is usually categorized as female or male but there is variation in the biological attributes that comprise sex and how those attributes are expressed. Gender refers to the socially constructed roles, behaviours, expressions and identities of girls, women, boys, men, and gender diverse people).

Line 216. Establishment of On-Ice Ventilation Threshold Heart Rate Speculation Formula: This still needs to be described prior in the Method section.

Conclusions

Please address comments below.

Specific comments

Line 343: “…can better evaluate the aerobic capacity of athletes skating on ice.” Compared to what? Better than what? Please clarify.

Author Response

Dear reviewer Thank you for your hard work. We benefit a lot from your comprehensive and detailed comments on our manuscripts. We have revised the article according to your comments. As for the language, we have invited relevant institutions and native speakers to help us polish it before submission, but we are very sorry that it has not reached your satisfaction. We have made targeted modifications as follows:
